# Wind turbine drivetrains: state-of-the-art technologies and future development trends

Amir R. Nejad[1], Jonathan Keller[2], Yi Guo[2], Shawn Sheng[2], Henk Polinder[3], Simon Watson[3], Jianning Dong[3], Zian Qin[3], Amir Ebrahimi[4], Ralf Schelenz[5], Francisco Gutiérrez Guzmán[6], Daniel Cornel[6], Reza Golafshan[6], Georg Jacobs[6], Bart Blockmans[7,8], Jelle Bosmans[7,8], Bert Pluymers[7,8], James Carroll[9], Sofia Koukoura[9], Edward Hart[9], Alasdair McDonald[10], Anand Natarajan[11], Jone Torsvik[12], Farid K. Moghadam[1], Pieter-Jan Daems[13], Timothy Verstraeten[13], Cédric Peeters[13], and Jan Helsen[13]

[1]Marine Technology Department, Norwegian University of Science & Technology, NO-7491, Trondheim, Norway
[2]National Renewable Energy Laboratory, Golden, CO 80401, USA
[3]Technische Universiteit Delft, Mekelweg 2, 2628 CD Delft, The Netherlands
[4]Leibniz University Hannover, Institute for Drive Systems and Power Electronics, Postfach 6009, 30060 Hannover, Germany
[5]Center for Wind Power Drives CWD, RWTH Aachen University, Campus-Boulevard 61, 52074 Aachen, Germany
[6]Institute for Machine Elements and Systems Engineering MSE, RWTH Aachen University, Schinkelstrasse 10, 52062 Aachen, Germany
[7]KU Leuven, Mechanical Engineering, Division LMSD, Heverlee, Belgium
[8]Flanders Make, Core Lab Dynamics of Mechanical and Mechatronic Systems, Heverlee, Belgium
[9]University of Strathclyde, 16 Richmond St, Glasgow G1 1XQ, United Kingdom
[10]Institute for Energy Systems, School of Engineering, Edinburgh, United Kingdom
[11]DTU Wind Energy, Frederiksborgvej 399, 4000 Roskilde, Denmark
[12]Equinor ASA, Sandslivegen 90, 5254 Sandsli, Norway
[13]Department of Mechanical Engineering, Vrije Universiteit Brussel / OWI-Lab, B-1050, Brussels, Belgium

**Correspondence:** Amir R. Nejad (Amir.Nejad@ntnu.no)

**Abstract.** This paper presents the state-of-the-art technologies and development trends of wind turbine drivetrains—the energy conversion systems transferring the kinetic energy of the wind to electrical energy—in different stages of their life cycle: design, manufacturing, installation, operation, lifetime extension, decommissioning, and recycling. Offshore development and digitalization are also a focal point in this study. The main aim of this article is to review the drivetrain technology development as well as to identify future challenges and research gaps. Drivetrain in this context includes the whole power conversion system: main bearing, shafts, gearbox, generator, and power converter. The paper discusses current design technologies for each component along with advantages and disadvantages. The discussion of the operation phase highlights the condition monitoring methods currently employed by the industry as well as emerging areas. This article also illustrates the multidisciplinary aspect of wind turbine drivetrains, which emphasizes the need for more interdisciplinary research and collaboration.

# 1 Introduction

The European Green Deal aims to make the European Union climate-neutral by 2050, with land-based and offshore wind being an important part to meet this target (EU, 2019, b). The European Union has been at the forefront of wind energy technology development in recent years, especially offshore—European companies represent an impressive 90% of the offshore global market (EU, 2019, a). There is a special focus on offshore wind development in the EU Clean Energy for All Europeans Package, in which 30% of the future electricity demand, approximately 450 gigawatts (GWs), is expected to be supplied by offshore wind (EU, 2019, a)—a huge increase from today's 20 GW of installed capacity (Wind Europe, 2020). In the United States, it has been estimated that wind can supply 35% of U.S. electricity demand by 2050, with 86 GW installed offshore (DOE, 2015). Moving from land-based to offshore turbines has also opened possibilities of increasing the size and power of the wind turbine and plant. Deeper water locations offshore have also been used by floating turbines, with the first floating wind plant in operation since 2017. Such fascinating developments are challenging the technological borders and existing knowledge base of the industry. There is limited experience with such huge machinery in harsh environmental conditions, so the best practices and standards have not yet fully matured.

The drivetrain converts mechanical to electrical power and transmits the rotor loads to the bedplate and tower. The drivetrain in this context includes the entire power conversion system from the main bearing to the electrical generator and power conversion system. The two main drivetrain configurations and components that characterize them are depicted in Figure 1. A variety of wind turbine drivetrain technologies are available, with pros and cons for each in terms of cost, weight, size, manufacturing, materials, efficiency, reliability, and operation and maintenance (O&M) (Polinder et al., 2006; Arabian-Hoseynabadi et al., 2010; Moghadam and Nejad, 2020; Harzendorf, 2021; Harzendorf et al., 2021). With digitalization expanding in all industries, new opportunities have arisen in the operation phase, including digital O&M and digital twins. As the age of the installed fleet continues to increase, considerations for lifetime extension and decommissioning are also becoming more important.

It is interesting to highlight that the technological drivers for the drivetrain are not necessarily the same as other elements of the wind turbine. For instance, for towers there are site-specific solutions that depend on wind conditions or other site characteristic, which affect the cost considerably. However for the drivetrain, the number or size of components are cost drivers, not the site or region-specific characteristics. It is possible to design the drivetrain with respect to logistic costs, for example a modular design that can be handled by a smaller or tower-top crane, reducing transportation, installation, and replacement costs for offshore wind turbines, although such technology is still under development. In terms of availability of future drivetrain designs, the development trend is not only on the quality and manufacturing, but also on the service and operational monitoring.

This study reviews the state of the art of the drivetrain technology in the wind turbine industry and discusses future development trends. The focus is on conventional and widely used concepts; unconventional designs, such as hydrostatic (Silva et al., 2014) and hydraulic designs, are not discussed. To achieve the aims of this paper, a life cycle approach (Torsvik et al., 2018), as illustrated in Figure 2, is employed. First, the design—and, to a limited extent, manufacturing—is discussed. It is followed by drivetrain operation—in particular, condition and performance monitoring; and, finally, lifetime extension, decommissioning, and recycling.

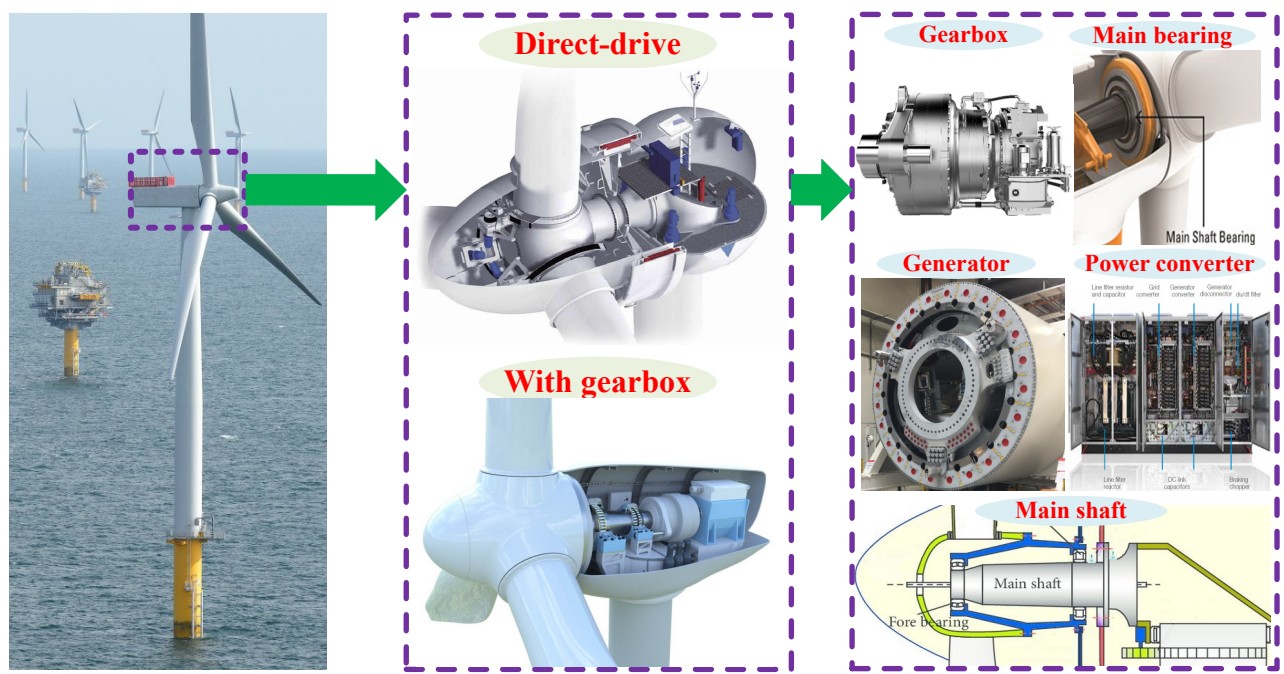

**Figure 1.** Drivetrain configurations and main components (photos and figures are adopted from Equinor; Schmid; OpenPR; Smalley (2015); Zheng et al. (2020); Siemens Gamesa; ZF WIND POWER; ABB).

## 2 Design trends and developments

A compact, lightweight drivetrain is the most cost-effective option for large offshore wind turbines because it reducues the nacelle mass and hence tower and foundation or floating platform masses and costs. To achieve these reductions, there has been a trend toward increasing the mechanical integration of the main bearing, gearbox, and generator (Stehouwer and van Zinderen, 2016; Demtröder et al., 2019; Nejad and Torsvik, 2021; Reisch, 2021; Zeichfüßl et al., 2021; Weber and Hansen, 2021). In terms of the power conversion system, permanent magnet synchronous generators (PMSGs) with full-power converter systems are becoming more common than doubly-fed induction generators (DFIGs) with partial-power converter systems. Concerns over the supply of rare-earth materials typically used in PMSGs have also spurred interest in alternate generator technologies, such as superconducting generators (Veers et al., 2020). Regardless of drivetrain design choice, the loads and operational conditions that the drivetrain is subjected to are derived from the design load cases described in the International Electrotechnical Commission (IEC) 61400-1 design standard for land-based wind and IEC 61400-3-1 and IEC 61400-3-2 design standards for offshore fixed and floating wind applications.

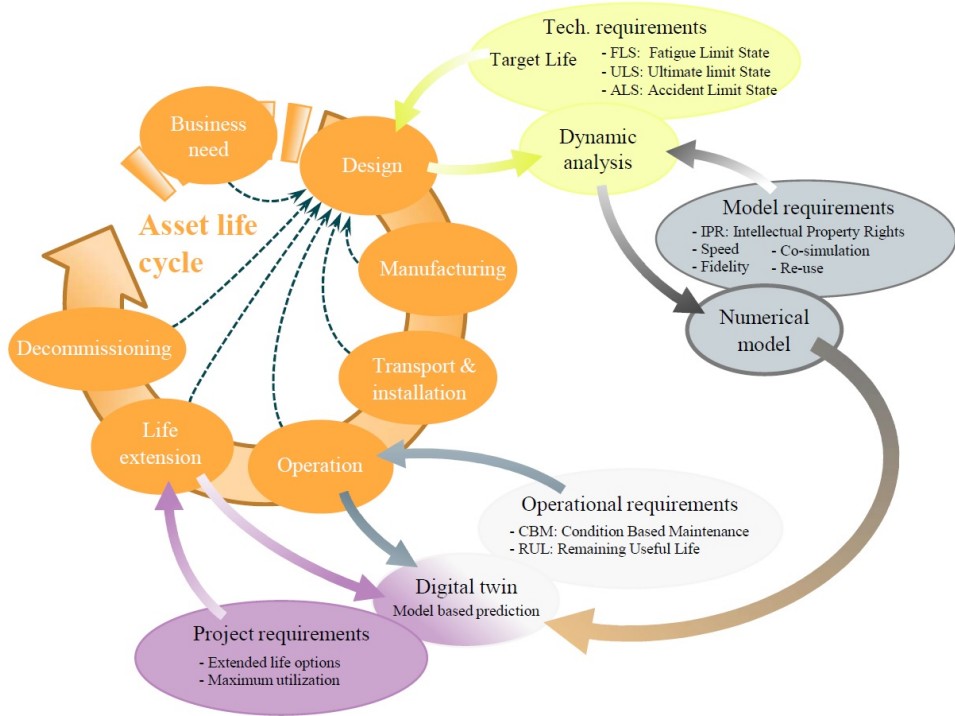

**Figure 2.** Wind turbine life cycle (Torsvik et al., 2018).

## 2.1 Main bearing

Current commercially available main bearing designs use rolling element bearings (Hart et al., 2020). Spherical roller bearings are utilised in a significant proportion of currently operational main bearings, with usage of this technology likely to continue (to a greater or lesser extent) in sub-5MW machines. Tapered roller bearing designs are now also common at these power levels. For larger wind turbines ($\geq$ 5MW), however, the industry has very much converged on tapered roller main bearing technology (Chovan and Fierro, 2021). Because of the large applied nontorque loads, the resulting bearing designs are likewise large in diameter, limited in size only by manufacturing and transportation restrictions. Continued use of rolling element bearings is likely because it is a familiar technology, albeit with design trends moving outside the envelope of prior experience. A central driver behind the move to large-diameter rolling element bearing arrangements is the need for cost-effective rotor support solutions. Other existing bearing technologies—such as hydrostatic, air, and magnetic bearings—tend to require very rigid support structures or are limited to smaller diameters than would be required by modern wind turbines, although hybrid solutions combining different technologies have been proposed (Shrestha et al., 2010). Given the continued increases in main bearing diameter, understanding the effects of deflections in large-diameter rolling element bearings is essential, and the current practice of assessing bearing design life through only the conventional calculation methods in the International Organization

for Standardization (ISO) standards 76 and 281 and technical specification (TS) 16281 might be insufficient with respect to the resulting service life observed in operation.

Main bearings have been shown to experience repeating, large-scale fluctuations in load, even during normal operation (Hart, 2020). These fluctuations likely increase the risk of other damaging mechanisms (such as roller skidding, surface fatigue, wear, and abrasion) not accounted for in fatigue-life calculations. As such, the analysis of the operating conditions of these components has further indicated that current life-assessment standards might be insufficient. Main bearing failure rates of up to 30% during a 20-year design life have also been reported (Hart et al., 2019). Further work is therefore needed to identify principal drivers of main bearing failures, allowing for the development of appropriate design standards and best practice specific to this component, which, in turn, will lead to improvements in reliability. Guo et al. (2021) investigated one possible driver of main bearing failures, that of axial motion between rollers and raceways. It has previously been hypothesised that such motions may compromise the lubricant film which separates bearing internal surfaces, leading to increased levels of friction and wear. However, from analysis of both field measurements and analytical model outputs it was concluded that axial velocities are too small to have any significant influence on lubricant film formation (Guo et al., 2021).

Modern offshore wind plants are high-value assets, and there is an increasing interest in longer design life and lifetime extension. Turbine size and drivetrain arrangement can, however, result in main bearing replacement becoming more difficult and expensive, generally requiring removal of the rotor. Consequently, main bearings will increasingly be regarded as part of the load-carrying structure, with cost implications of failure more severe as a result. A further ramification of increased levels of integration is that main bearing operational requirements become linked to those of other components. For example, in addition to supporting the turbine rotor, some direct-drive configurations require the main bearing to also support the generator rotor while maintaining an appropriate generator air gap. Coupled approaches to the modeling and assessment of wind turbine drivetrain systems will therefore become increasingly important.

Novel main bearing design concepts are also being developed and tested. Loriemi et al. (2021) propose the use of asymmetric spherical roller bearings to improve main bearing internal load sharing during operation. Finite element modelling was used to compare a standard and asymmetrical design, with results indicating significant improvements for the latter with respect to both main bearing fatigue life and isolation of the gearbox from transferred axial loads. Plain bearing (equivalently, journal bearing) technology is also being considered in this space. More specifically, a segmented plain bearing with conical sliding surfaces is being developed as a main bearing solution for wind turbines (Rolink et al., 2020, 2021). Sliding segments are connected to the housing via doubly flexible supports to maintain their alignment with the tilting shaft and prevent edge loading. A major advantage of this design is that the sliding segments are individually replaceable uptower, without requiring drivetrain disassembly. Other main bearing concepts utilising this same technology have also been explored (Rolink et al., 2020).

With respect to materials, chill-cast nodular cast iron (GJS) has been investigated as a possible alternative to forged steel for manufacture of the rotor-shaft/main bearing seat (Kirsch and Kyling, 2021). Chill-cast GJS enables lightweight construction through freedom of design, while also providing appropriately robust mechanical properties. Lighter and smaller hollow rotor-shaft designs may become feasible as a result. This, in turn, would have implications for main bearing design/selection. Cost reductions may therefore be possible through design optimisation of GJS components, but, resulting shaft-bearing systems can

have reduced stiffnesses which may lead to inner ring creep or fretting fatigue (Kirsch and Kyling, 2021). Work in this area is ongoing.

## 2.2  Gearbox

Wind turbine gearboxes continue to increase in size (up to 3 m in diameter) and power (up to 15 megawatts (MW)) (Vaes
et al., 2021). With multistage gearboxes using four or more planets per stage, torque densities of 200 newton-meters per kilogram and speed increasing ratios up to 200 are now available (Daners and Nickel, 2021). These increasingly flexible systems require detailed modeling to understand the effect of deformations and dynamics on internal loading (Wang et al., 2020). To achieve further cost reductions through economies of scale, modular gearbox designs have been introduced (Windpower, 2021). Gearboxes are designed for a minimum of a 20-year life, as specified in the IEC 61400-4 and American Gear Manufacturers
Association (AGMA) 6006 gearbox design standards. Provisions for up-tower service or replacement of gearbox components is becoming more common and is required for components that have a design life less than the gearbox. The gearbox system comprises many elements (primarily the rotating shafts, gears, and bearings), so the reliability of the gearbox is the product of the reliability of all the failure modes for which there exists a reliability calculation as described in Verband Deutscher Maschinen- und Anlagenbau 23904 and IEC Technical Specification 61400-4-1. But many, if not most, of the failure modes
experienced in operation do not have a standardized reliability calculation; hence, as described earlier, there exists a difference between the apparent reliability observed in operation and the calculated design reference reliability. This is not unusual, and it occurs in other industries, although the O&M cost impact for wind turbines can be more severe. For gearboxes, the reliability calculation considers gear tooth surface durability (pitting) according to ISO 6336-2 and bending strength according to ISO 6336-3, rolling element bearing rating life from subsurface-initiated fatigue (i.e., rolling contact fatigue) according to ISO 281
and ISO/TS 16281, and shaft fatigue fracture according to Deutsches Institut für Normung 743 and American National Standards Institute (ANSI)/AGMA 6001. In some cases, a safety factor for or percentage risk of these failure modes can at least be quantified in the gearbox design process, including gear tooth scuffing according to ISO/TS 6336-20, ANSI/AGMA 925, and ISO/TS 6336-21; and gear tooth micropitting according to ISO/TS 6336-22; or otherwise assessed for gear tooth flank fracture according to ISO/TS 6336-4. Safety factors for the static strength of gears and bearings is calculated according to
ISO 6336 and ISO 76, respectively. Other bearing failure modes, such as surface-initiated fatigue (e.g., micropitting), adhesive wear, corrosion, electrical damage, and white-etching cracks can only be assessed qualitatively. Requirements for materials, processing, and manufacture are part of these standards. Further in-depth contact and finite element analysis are used to design the microgeometry of these rotating components, provide additional rating life calculations (Morales-Espejel and Gabelli, 2017), and analyze the supporting housing structures. In addition to classical reliability approaches, use of structural reliability
methods for reliability analysis of gears has also been investigated (Nejad et al., 2014a; Dong et al., 2020). Design guidance for the use of plain bearings in the gearbox is under development because they offer advantages in terms of torque density and are life-limited only by wear rather than determined by load-dependent rolling contact fatigue, although they are already becoming common in new gearboxes (Weber and Hansen, 2021; Zeichfüßl et al., 2021). Surface engineering, lubricants, and lubrication

of the gearbox also play an essential role in gearbox design, operation, and reliability (Dhanola and Garg, 2020; Jensen et al., 2021).

## 2.3 Generator

As highlighted earlier, wind turbine drivetrains can be either geared or direct-drive generator systems (Polinder et al., 2013). The geared generator system can be further divided into either a DFIG with partial power converter or a brushless generator with full power converter (GFPC) system. The DFIG system has been the most popular topology for medium-size turbines ranging from 3-6 MW. The GFPC system uses either a squirrel cage induction generator or a PMSG. Many manufacturers now provide commercial GFPC solutions at power levels up to 10 MW (Siemens, 2020; ABB, 2020). In terms of direct-drive systems, rare-earth PMSGs are appealing for offshore applications. The mainstream power level is from 5–7 MW, but the top power level has kept increasing during the past two decades.

The stator elements of DFIGs and PMSGs are largely the same. The major difference in terms of the electrical machine hardware is the rotor design and the means—-or lack thereof—-of getting current onto and off the rotor. These differences can impact both efficiency and failure mechanisms and their rates. In terms of efficiency, induction generators use a set of currents on the rotor to produce the rotor magnetic field. This leads to Joule rotor losses and hence a decrease in efficiency. In contrast, a PMSG uses rare-earth permanent magnets to produce the rotor magnetic field, hence avoiding further Joule losses.

In terms of failure types, a DFIG uses carbon brushes and slip rings to conduct the currents between the rotor and the stator. The brushes typically wear out over time and need frequent inspection and replacement. The PMSG avoids those elements. There are also differences in reliability because of the presence of conductor and insulation systems (DFIG) and magnet materials (PMSG), but those are not yet clear. A comparative study of DFIGs and PMSGs showed that during the early life, a PMSG has a failure rate 40% lower than that of a comparable DFIG (Carroll et al., 2014).

Alongside this variation in the reliability of different electrical machine architectures, there is variation in the reliability caused by the torque rating of the generator. This was first shown by Spinato et al. (2009). For example, it is possible to conceive of two wind turbines that both use the same generator type, but one is in a geared configuration (with a gearbox ratio of 100), and the other is direct drive. For the same wind turbine rotor (and subsequent power and rotational speed), the torque rating of the direct-drive generator will be 100 times more than that of the geared generator. Assuming that the same electromagnetic shear stress is produced by the two generators, the volume of the direct-drive generator will also be 100 times that of the higher speed generator. If they have the same ratio of diameter to axial length, then the generator diameter will be $\times\sqrt[3]{100}$ (i.e., ×4.64) that of the geared machine. The electromagnetic materials are approximately proportional to the surface area of the rotor and stator. In the case of the low-speed machine, this might be $\times 4.64^2$ (i.e., 21.5) times that of the higher speed machine. With more poles, more coils, longer conductors, and insulation, it is likely that the failure rate is higher in direct-drive machines if there is no improvement in failure rate intensity. These variations in generator failure rate should be taken into consideration - along with the gearbox failures discussed in Section 2.2 - when assessing failure rates of different types of geared and non-geared drivetrains.

The reliability and availability of the wind generator system has a decisive impact on the cost of energy (COE), especially offshore (Carroll, 2016; Shipurkar et al., 2016). The generator design should consider the interactions with other components to improve the system reliability of the drivetrain (Moghadam and Nejad, 2020). In large wind turbines, multiphase windings with modular converters can be used to improve the generator system availability (Shipurkar et al., 2015; McDonald and Jimmy, 2016).

Upscaling is still a continuing trend for both land-based and offshore wind turbines because a higher power wind turbine system leads to a lower Levelized Cost of Energy (LCOE) (Sieros et al., 2012). For land-based wind, recent developments include the design of 6-MW and 8-MW turbines, which will be on the market in the near future, whereas offshore commercial applications now aim for 10–15 MW, and research is going beyond 15 MW (Gaertner et al., 2020; Ashuri et al., 2016; Sartori et al., 2018). Upscaling brings many challenges, including large generator weight, high manufacturing/installation difficulties and cost, complicated electromechanical dynamics, and complexity of system monitoring. A systematic design approach will be required for the design of the generator, where cooling and efficiency will be among the challenges in higher power. Depending on the type, nominal power, shaft speed, the specific electric loading, and subsequently the armature thermal loading, three general solutions—namely, air-air, air-water, and water jackets—are commercially available to implement the cooling system of a wind generator (Polikarpova et al., 2014). Significant cost savings can be realized with the development of a more effective stator winding cooling system that further limits the current density to enable the development of higher power PMSGs of substantially smaller diameters while not adversely affecting the electromagnetic performance of the generator.

Multiphase, modular designs are solutions to tackle some of the challenges, and they have been used in commercial systems (Yaramasu et al., 2015; McDonald and Bhuiyan, 2016). Concerns over the availability of rare-earth elements—such as neodymium, praseodymium, and dysprosium—typically used in PMSGs have led the wind and other industries to develop innovative technologies to reduce, substitute for, or entirely eliminate their need in generators (Veers et al., 2020). This also results in technologies that are lighter than PMSGs. Breakthroughs in superconducting materials could change the scenario of materials and upscaling completely (Hoang et al., 2018), with several superconducting generators successfully tested (Frank et al., 2003; Bergen et al., 2019) or in development (Moore, 2020).

The interactions between the generator and power electronics can bring issues including bearing currents, additional stress in insulation because of overvoltage in transients, and high-voltage slew rates (Chen et al., 2020); therefore, these interactions should be studied and modeled not only for the design but also for O&M. Proper filters and control methods should be integrated according to the generator types and power electronics topologies. For the upscaling of wind generators, various modular and multilevel power converter topologies feeding multiphase windings will be a promising solution. But attention should be paid to circulating current and potential asymmetric supplies to avoid risks (Yaramasu et al., 2015).

## 2.4 Power converter

The power converter is responsible for controlling the output power of the generator with regulated voltage and frequency (Moghadam et al., 2018). Wind turbine power converters used to have a topology as shown in Figure 3, top, where the generator-side converter is a diode rectifier cascaded with a boost converter to maintain a stable direct current (DC) link voltage, then a

two-level inverter is employed on the grid side to ensure full control of the grid current injection (e.g., total harmonic distortion and power factor). This topology has a relatively low cost because of fewer power switches than newer configurations, so it is widely used for generators in small- to medium-size wind turbines. For megawatt-scale generators, however, the low-frequency torque pulsation and high total harmonic distortion become very harmful to the generator. As a result, the design of the boost

converter becomes very challenging, and therefore the front end is then replaced by a two-level, six-switch converter with power factor correction, which is shown in Figure 3, bottom, and is called a back-to-back (BTB) converter. A DFIG with partially loaded BTB converter is commonly used for generators less than 3 MW, whereas a PMSG with a fully loaded BTB converter is commonly used for generators greater than 3 MW.

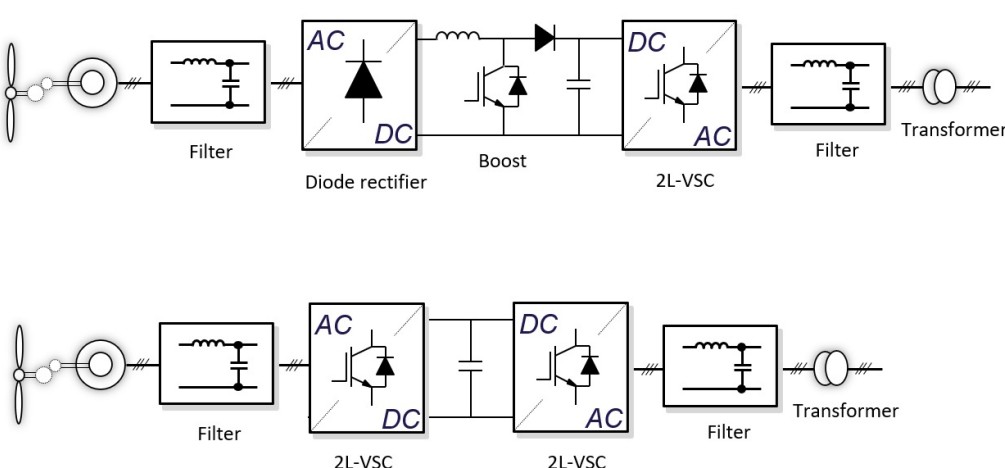

**Figure 3.** Typical wind power converter topologies from the kilowatt to megawatt scale (Blaabjerg and Ma, 2013). Top: Diode rectifier + boost DC/DC + two-level voltage source converter (VSC). Bottom: Two-level back-to-back converter.

When the power rating is 10 MW or more, a single two-level BTB converter is no longer suitable because the current stress

of the power devices would be extremely high. For two-level BTB converters, the grid-side voltage is typically 690 V. To solve this, multiple two-level BTB converters can be connected in parallel to share the current, whereas the connection of the generator side can be slightly different depending on whether the generator has a multiwinding (see Figure 4, top). Another way to upscale the power rating is to increase it to medium voltage, for instance, by a neutral-point-clamped (NPC) converter (see Figure 4, down) or a modular multilevel converter.

Other crucial topics for power converters include thermal loading and grid integration. Intermittent winds create temperature swings in power converters, which are the main factor of power converter aging. The possibility of applying Lithium-ion battery energy storage systems to smooth wind power has been investigated to reduce this aging (Qin et al., 2013). The approach is effective, and the stress mitigation performance is affected by the energy and power rating of the energy storage system; however, the energy storage system's high cost is still the main barrier preventing its widespread application. Variable switch-

ing frequencies can also somewhat reduce temperature swings (Qin et al., 2015b), which is an attractive option as a control

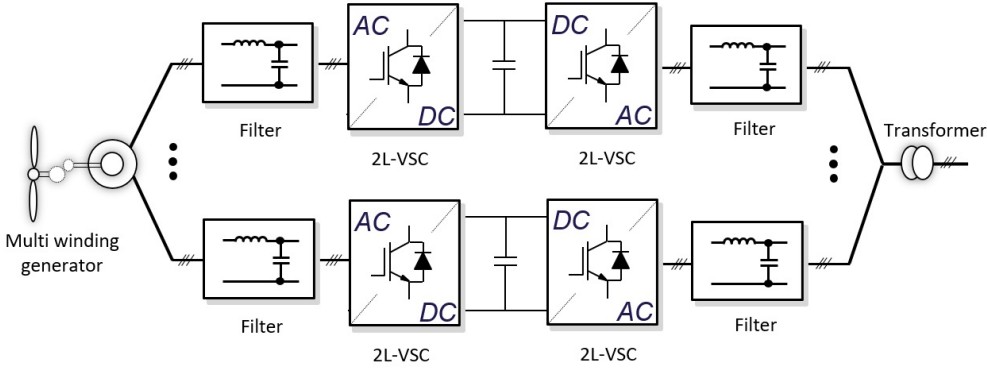

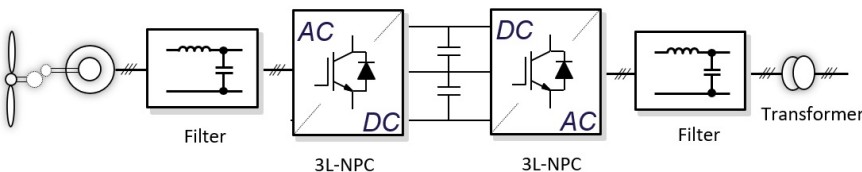

**Figure 4.** Beyond 10-MW wind power converter topologies. Top: Two-level BTB VSC in parallel. Bottom: Three-level NPC converter.

approach that does not add hardware cost. Nonetheless, the grid filter design to handle the variable switching frequency might be challenging. A promising approach can be using the kinetic energy in the wind turbine's rotor as energy storage by the rotating speed control to suppress the power fluctuation in the power converter and thereby reduce the temperature swings (Qin et al., 2015a). Another topic that is trending for the wind power converter is grid integration. Grid-following mode control

5 was applied in the grid-side converter, and it is still mainstream; however, as wind power penetration is increasing, the grid is becoming relatively weak (low short-circuit ratio, low inertia, etc.). Grid-supporting mode control can cause power quality issues and even grid failures in some severe cases (Larumbe et al., 2018, 2019). Energy storage systems or synchronous condensers can be associated with wind plants to provide inertia and to reduce the burden of the grid without grid reinforcement (which is very expensive), but these components are still expensive. Another promising approach is to apply grid-forming mode

10 control to the grid-side wind power converter, so the inertia in the wind turbines can be used to support the grid and enhance grid stability and reliability. Grid-forming control is a group of controllers with which the wind power converters not only get the grid current controlled, but also support the grid voltage, frequency, or inertia. Typical grid-forming control includes droop control, in which the active/reactive power injected to the grid is reacting to the grid frequency/voltage, and virtual synchronous machine control, in which the inertia of the synchronous machine is mocked in wind power converter.

15     There is a trend in power electronics to evolve from silicon-based power semiconductors to wide-bandgap devices (e.g., silicon-carbide devices). This will have a positive impact on wind energy systems because it can improve the power den-

sity and improve the efficiency of the power converters (Erdman et al., 2015). In the meantime, it also brings challenges to wind generators because of the high-voltage slew rate as a result of fast switching. Proper filtering and oscillation damping technologies should be used to mitigate the side effects, such as common mode current and insulation degradation.

The type of cooling system chosen for the converter depends on the nominal power and voltage, power density and thermal design, and generator technology, which can be based on either air or direct/indirect liquid cooling (Zhou et al., 2013). By choosing a liquid cooling system, the size of the converter for high-power applications (>5 MW) can be significantly reduced.

## 2.5 Modeling and analysis

Wind turbine drivetrains are subject to dynamic loading from a wide range of operating conditions caused by wind shear, veer, turbulence, and gusts; changes in the turbine operational state; grid faults; and nacelle motions; therefore, it is essential that the computational models for the drivetrain consider the dynamics of the rotor and the demands of the grid through the converter. Such an electromechanical model captures the aeroelastic interactions of the rotor and characterizes the voltage/current excursions in the generator because of grid requirements (Gallego-Calderon et al., 2017; Bruce et al., 2015; Blockmans et al., 2013).

Given the complex contact mechanics, one of the most challenging aspects of simulating wind turbine gearboxes is modeling the meshing gear teeth and supporting bearings. With their computational efficiency, lumped-parameter methods have been the method of choice for modeling gears and bearings in system-level, flexible, multibody systems. In this approach, a pair of meshing gears is simplified into a pair of rotating cylinders with constant inertias that are interconnected by a (nonlinear and/or time-varying) spring along the line of action or contact. Analogously, in lumped-parameter bearing modeling, the contact of the rolling elements with the inner and outer bearing raceways are described in terms of nonlinear springs. For meshing gear teeth, the tooth stiffness changes periodically as the number of teeth in contact and the contact locations along the active tooth flanks change throughout the gear rotation, whereas for rolling element bearings, the stiffnesses vary with the magnitude and the location of the rolling element contact loads. The various lumped-parameter models differ primarily in the way the contact is computed. One of the oldest but most complete approaches for modeling the interactions between contact surfaces is the classic contact theory by Hertz and its derivatives (Johnson and Johnson, 1987). Hertzian contact theory is valid i) when the contact area is sufficiently far from the boundaries of the contacting bodies, allowing them to be treated as elastic half spaces; and ii) when the elastic deformation of the body is confined to the contact zone. These assumptions have proven to be particularly valid approximations in bearing analysis, where the direct application of Hertzian contact theory has lead to computationally efficient and accurate three-dimensional ball bearings (De Mul et al., 1989a; Lim and Singh, 1990) (see Figure 5a). For gears, however, the Hertzian assumptions are not consistent with the comparatively large bending, compressive, and shear deformations that are typically encountered in meshing gears. Lumped-parameter gear modeling techniques are therefore typically formulated using mesh stiffnesses that are obtained through empirical formulations (Cai and Hayashi, 1994), analytic techniques based on a combination of linear elasticity theory and Hertzian contact theory (zu Braunschweig. Institut für Maschinenelemente, 1951; Wang et al., 2018), or polynomial curve fits that are derived from finite element simulations (Kuang and Yang, 1992). Note that these mesh stiffnesses are generally formulated per unit of face width and thus assume a uniform contact force distribution along

the face width. In helical gears, however, the contact forces are nonuniformly distributed across the tooth face width, depending on the gear microgeometry, the helix angle, and possible gear misalignments. Although three-dimensional lumped-parameter models have been formulated for helical gears (Eritenel and Parker, 2012), a more common approach in wind turbine gearbox modeling—as is also available in a number of commercial, flexible, multibody simulation software packages—is to divide

each helical gear into a number spur gear slices and to sum the stiffnesses and/or contact forces of the individual slices (Feng et al., 2018) (see Figure 5b). A similar slicing approach is applied in lumped-parameter roller bearing modeling, where the contact force distribution over the roller surfaces is a nonuniform line load (De Mul et al., 1989b). Note that although lumped-parameter approaches yield reasonably accurate and efficient ball and roller bearing models, contrary to lumped-parameter gear contact models, these bearing models are rarely directly integrated into system-level wind turbine drivetrain models; instead,

the models are used to derive linearized stiffness matrices that describe the three-dimensional behavior of the bearing at a selected operating point (Helsen et al., 2011).

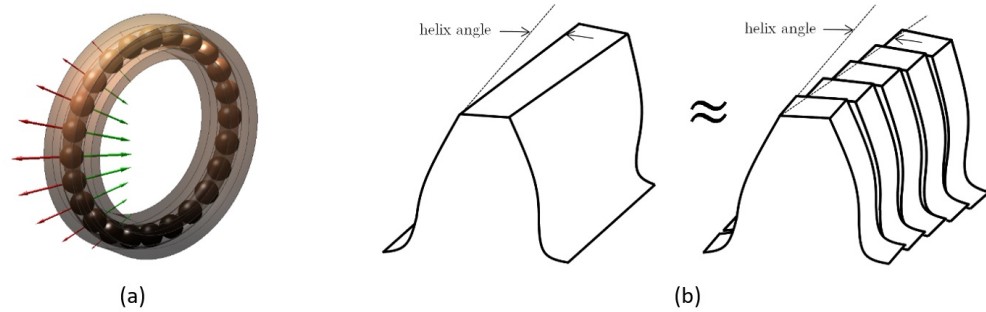

**Figure 5.** (a) Lumped-parameter bearing model based on Hertzian contact theory; (b) helical gear slicing for three-dimensional lumped-parameter gear analysis.

Although lumped-parameter models enable the rapid construction and efficient evaluation of gear and bearing models, they lack the modeling complexity required to evaluate dynamic behavior with, for example, gear geometric modifications or housing flexibility. This is desirable, e.g., in the analysis of planetary gear sets, where the flexibility of the ring-housing assembly

has a considerable impact on the overall gearbox behavior (Hu et al., 2019). In distributed parameter methods, such as the finite element method, the full geometric extent of the gear pair is considered, whereas a large part of the lumped-parameter assumptions are replaced by first principles. The increased accuracy of these methods comes at the price of an increased computational cost that is compatible only with static simulations (e.g., computing a linear bearing stiffness matrix (Guo and Parker, 2012) or the stiffness maps of a pair of gears (Palermo et al., 2013)). To alleviate the computational burden of the finite element

method in dynamic simulations, two approaches have recently been introduced. The first approach combines the finite element method with semi-analytic results from classic contact theory to eliminate the need for highly refined finite element meshes in the zone of contact (Andersson and Vedmar, 2003; Vijayakar, 1991). The second approach reduces the number of degrees of freedom in the finite element models by applying model order reduction techniques that are specifically tailored toward

dynamic contact problems (Blockmans et al., 2015; Fiszer et al., 2016). Given the complexity of these methods, however, the usage of finite element method-based techniques to model gears and bearings in system-level drivetrain simulations remains largely restricted to the preprocessing phase of the simulation or limited to static simulations. In the absence of complex contact interactions, the modally condensed finite element method (Craig Jr and Ni, 1989) becomes a practical means for modeling

complex, flexible components such as planet carriers, housings, and the bedplate that exhibit relatively low-frequency modal behavior; whereas shafts are commonly represented by Timoshenko beam elements (Struggl et al., 2015). With the increasing size of wind turbines, the flexibility of these components becomes increasingly important because it can significantly affect internal load distributions and vibrations (Helsen et al., 2012). Components with high stiffness-to-mass ratios, on the other hand, are modeled as rigid bodies in which the number of degrees of freedom equals six minus the number of applied mo-

tion constraints. Couplings such as universal and revolute joints are typically represented by algebraic constraint equations, whereas interference fits, spline couplings, and bolted connections are commonly idealized into perfectly homogeneous rigid connections (Marrant et al., 2010). Another modeling approach for spline couplings is to consider rigid in rotation but soft in tilting directions (Guo et al., 2016).

     Although significant strides have been made in recent years to increase the accuracy of wind turbine gearbox simulations,

a number of challenges remain. With regard to gear and bearing simulation, the modeling of contact damping phenomena is not nearly as effective and well understood as the modeling of stiffness-related effects despite meritorious contributions in this direction (Li and Kahraman, 2013). In addition, the same techniques that resulted in effective lumped-parameter gear contact models have failed to achieve similar results in the field of spline modeling. This is largely because of the relatively large dimensions of the contact zone, rendering Hertzian techniques inaccurate and slicing techniques impractical. With unreduced

finite element-based approaches (Kahn-Jetter and and Wright, 2000) as the main resort, simulations often idealize spline couplings, which can significantly impact the contact load distributions, especially in planetary gear stages. Finally, with the number of gearbox modeling approaches continuously increasing in both the scientific literature and in commercial software, there is a need for identifying and validating the required levels of modeling accuracy in gearbox analyses, including forward dynamic analyses (He et al., 2019), durability analyses (Ding et al., 2018), transfer path analyses (Vanhollebeke et al., 2015),

and inverse analyses (Bosmans et al., 2020).

     Wind turbine generators can be modelled in different levels of details, depending on the purpose of the study (Asmine et al., 2010; Ugalde-Loo et al., 2012). When power grid interaction of the wind plants are studied in which the performance of individual wind turbines are not concerned, the individual generators can be lumped into equivalent machines represented at the collector buses (Kazachkov et al., 2003). To evaluate the impact of wind turbines on the power grid by simulating power sys-

tem dynamics, many simplified approaches have been implemented depending on the level of detail required (Slootweg et al., 2003; Akhmatov et al., 2003; Ekanayake et al., 2003; Ullah et al., 2008). For power system design, power flow calculation, short circuit modelling and power stability analysis, usually positive-sequence or RMS models are used, in which electromagnetic transients are neglected (Asmine et al., 2010). In IEC 61400-27-1:2020, standard electrical simulation models of wind turbine generators for power system and grid stability analysis are described (Göksu et al., 2016). To address electromagnetic

transients, e.g. when fault ride through strategies are considered, electrical generator models should be built based on their

voltage equations (Trevisan et al., 2018). Constant lumped parameters are typically used in these equations. For design and optimisation of wind turbine generators, detailed electromagnetic models based on either analytical methods or finite element methods (FEM) are necessary. For final design and optimisation, FEM is extensively used (Duan et al., 2010). For detailed analysis of generators under transients, e.g. in a fault run through, circuit-coupled FEM can be used to consider the parameter variations caused by electromagnetic nonlinearity. Cooling design of generators requires accurate calculation of losses and temperature rises, which is usually carried out by coupled FEM or thermal circuit analysis (Bhuiyan and McDonald, 2018; Kowal et al., 2013). As wind turbines grow in size, the electromagneto-mechanical dynamics of the generator and its associated mechanical components should be analysed in detail to avoid undesirable vibrations. In these analyses, three-dimensional FEM is sometimes necessary to consider the rotor skewing. The whole machine must be modelled to consider the eccentricity by estimating the radial deformation. Electromagnetic design of large wind generators should consider both the power electronics and mechanical interactions thoroughly, and pole-slot number combination, air-gap length, and air gap radial forces, cogging torque and iron saturations, Eigen modes and Eigen frequencies, should be carefully studied in the electromagneto-mechanical coupling analysis to result in a robust design (Sopanen et al., 2010; Kirschneck et al., 2015; Desmedt et al., 2020).

Similarly, wind power converters can also be modelled from different aspects with different level of details. Power loss and thermal models are typically used for selection of power switches, design of the cooling system, and reliability studies (Qin et al., 2015b), which is no different from power electronics in other applications. For controller design and ensuring the operation stability, electrical dynamic modelling of the wind power converter is needed, in which the coupled model of plant, PWM (pulse-width modulation) converter, power filters and power grid is engaged. In earlier stages of control design, the average model of PWM converter is used to support the less computationally intensive analysis of the controller performance, whereas in later stages the full switching model is replaced. For grid impact study, the impedance model of the grid-side wind power converter is usually applied. In the impedance model, the wind power converter is typically modelled as a Norton equivalent, composed of a current source and an output impedance (Larumbe et al., 2021; Vree et al., 2020). The impedance model is especially appropriate for studying the harmonics or stability of a wind plant, because the impedance model of a single wind power converter can easily be aggregated into a plant. Not only the power filter but also the control of the wind power converter affects its impedance model, including the current controller, phase lock loop, and even the DC link voltage controller.

## 3   Operation and maintenance

O&M costs represent a sizable and potentially increasing share of the LCOE, especially as wind's LCOE declines because of reduced upfront costs and improved performance. Recent data suggest that O&M can account for around 25% of the LCOE for onshore installations (Renewable Energy Agency, 2012; Wiser et al., 2019; Steffen et al., 2020; Ren et al., 2021) to more than 35% for offshore installations (Carroll et al., 2017; Röckmann et al., 2017; Ren et al., 2021). Downtime must also be considered. Additionally, increased availability might not lead to reduced O&M costs offshore because vessel type costs must also be considered for turbine repair. For example, crew transfer vessels have much lower costs than jack-up vessels.

The maintenance paradigm is shifting from reactive, periodic time or usage-based maintenance to condition, reliability-centered (or predictive) maintenance supported by digital twins as elaborated in following section 6.3. If the turbines do not have dedicated condition monitoring systems, some anomaly detection or fault diagnosis can be conducted based on the turbine supervisory control and data acquisition (SCADA) system data, for which the main purpose is operational control.

## 3.1 Condition monitoring and fault detection

Condition monitoring is an umbrella term that spans various different ways of tracking the health state of a machine in which typically vibration, temperatures, oil contamination, or electrical signatures from the generator are used as input signals (Fu et al., 2017; Qiao and Qu, 2018). By using appropriate analysis methods, system changes caused by damaged components (e.g., flaking of bearing raceways) or faulty system states (e.g., water contamination) can be identified. A recent review of the condition monitoring of drivetrains is offered by Helsen (2021). The following sections discuss the two most commonly used types of condition monitoring for wind turbine drivetrains along with the acoustic emissions approach.

### 3.1.1 SCADA-based condition monitoring

Wind turbine SCADA systems produce hundreds of channels of data concerning the operation of a turbine based on multiple installed sensors installed. In reality, only a small fraction of these data provide valid information that can be used for condition monitoring, and there is a significant challenge in how to extract which information is important. SCADA data in wind turbines are typically sampled at around 1 Hz and averaged every 10 minutes. From a condition monitoring point of view, these data can serve as a low-cost potential solution because no extra sensors are required. The entire list of parameters tracked in the SCADA data is typically quite extensive, but an overview of basic SCADA parameters is given in Table 1. Normally, the extent and quality of the SCADA data depend on the turbine manufacturer, though IEC 61400-25 is used as a common protocol for data collection and labelling of parameters. Other possible uses besides condition monitoring include power curve analysis (Lydia et al., 2014) and modeling with, e.g., k-nearest neighbors (Kusiak et al., 2009), spare part demand forecasting (Tracht et al., 2013), and load monitoring (Wächter et al., 2015). Angular velocity measurements from SCADA have also been used for fault detection and remaining useful life (RUL) (Nejad et al., 2014b, 2018; Moghadam and Nejad, 2021, 2022).

To analyze SCADA data, machine learning techniques are often employed. One classification of machine learning techniques for wind turbine condition monitoring is to divide them into supervised (classification and regression) and unsupervised (clustering) learning. An example of classification for fault detection, isolation, and failure mode diagnosis on the gearbox is illustrated by Koukoura et al. (2019), whereas Turnbull et al. (2019) applied a combination of clustering and classification techniques to group similar operating conditions and detect generator faults. An example of regression to detect anomalies in vibration indicators can be found in Verstraeten et al. (2019). The authors use Bayesian ridge regression to fit linear parameters and inherent noise to the observed data while maintaining the uncertainty over the parameters. This way the model can distinguish between expected and anomalous behavior while capturing the stochasticity of the parameters. In Helsen et al. (2018), an ensemble of models is used to classify bearing temperature data in normal and anomalous behavior. An extensive review of machine learning approaches applied to wind turbines can be found in Stetco et al. (2019).

**Table 1.** Overview of some basic SCADA parameters (based on (Tautz-Weinert and Watson, 2016; Yang et al., 2013, 2014; Godwin and Matthews, 2013; Garcia et al., 2006; Zaher et al., 2009; Catmull, 2011; Watson et al., 2011; Schlechtingen et al., 2013; Wilkinson et al., 2014; Sun et al., 2016)).

| Category | SCADA parameter |
| --- | --- |
| Environmental | Wind speed, wind direction, ambient temperature, nacelle temperature |
| Electrical | Active power output, power factor, reactive power, generator voltages, generator phase current, voltage frequency |
| Control variables | Pitch angle, yaw angle, rotor shaft speed, fan speed/status, generator speed, cooling pump status, number of yaw movements, set pitch angle/deviation, number of starts/stops, operational status code |
| Temperatures | Gearbox bearing, gearbox lubricant oil, generator winding, generator bearing, main bearing, rotor shaft, generator shaft, generator slip ring, inverter phase, converter cooling water, transformer phase, hub controller, top controller, converter, controller, grid busbar |

Other researchers, for example, Tautz-Weinert and Watson (2016), categorize the different SCADA-based monitoring methods into five classes: trending, clustering, normal behavior modeling (NBM), damage modeling, and assessment of alarms and expert systems.

*i. Trending*    A very straightforward approach is to monitor the SCADA parameters over a long period of time and to use statistical thresholds for alarming.

*ii. Clustering*    When large numbers of wind turbines need to be monitored efficiently, it becomes imperative to have an automatic manner to classify the turbines as "healthy" or "faulty." Examples of clustering can be found in Kusiak and Zhang (2010), in which drivetrain and tower accelerations are analyzed using SCADA data by means of a modified k-means clustering conditioned on the wind speed. Kim et al. (2011) and Catmull (2011) applied self-organizing maps instead of k-means to build such clusters. Despite these advancements, the interpretation of the clustering results is often still perceived to be difficult (Tautz-Weinert and Watson, 2016).

*iii. NBM*    NBM employs the same idea of anomaly detection as the previous techniques, but it focuses more on the empirical modeling of the measurements. The residual error between the modeled and observed parameter then serves as a health indicator. A basic example of NBM involves the use of linear and polynomial models. A linear autoregressive model with exogenous inputs was used by Garlick et al. (2009) to detect generator bearing failures from the bearing temperature. Higher-order polynomial full signal reconstruction models of drivetrain temperatures were developed by Wilkinson et al. (2014) to detect gearbox and generator failures. A popular form of NBM is to track changes in the power curve as a function of wind speed. Faults such as problems with blade pitching can change the shape of the curve, though how to detect changes in the power curve, which can show a significant degree of variation even under normal conditions, is a challenge. A method which has shown promise is to use a Gaussian process (GP) model, *e.g.*, (Zhou et al., 2014; Manobel et al., 2018; Pandit and

Infield, 2018b). Such a model is able to fit an accurate power curve to noisy data which can then be used to predict future power values against which observed values can be compared. Pandit et al. (2019) showed that multivariate input GP models incorporating both wind speed and air density were especially suitable for this purpose. In addition to power, GP models can also be used to monitor other parameters such as blade pitch angle and rotational speed (Pandit and Infield, 2018a).

*iv. Damage modeling*      Instead of training empirical normal behavior models, the measurements can be interpreted with physical models to improve the accuracy. Gray and Watson (2010) developed a damage model using physical failure modes of interest to estimate the failure probability. A general scheme for a physics-based monitoring approach was proposed by Breteler et al. (2015) .

     *v. Assessment of alarms and expert systems*      The last class looks at the outputs of the SCADA control alarms or the NBM

output alarms. A typical example of this class is the analysis of the status codes of the wind turbine. Status code processing approaches typically investigate the possibility of extracting useful, actionable information about the health of the turbine from these status codes, and there exist many different ways to do this. For example, Chen et al. (2012) used a probabilistic approach with Bayesian networks to track down root causes for failures such as a pitch fault. Qiu et al. (2012) also used Bayes' theorem and compared the extracted patterns using a Venn diagram. Other approaches often involve machine learning, such as Kusiak

and Li (2011), who used neural network ensembles to predict status codes and their severity to detect a malfunction. Last, a significant amount of research examines the use of "expert" systems to interpret the status codes or model outputs. Often these systems are based on using fuzzy logic to determine a diagnosis for anomalies. Example research works that are based on or employ fuzzy logic are given in (Garcia et al., 2006; Schlechtingen et al., 2013; Sun et al., 2016; Cross and Ma, 2015; Li et al., 2013, 2014).

**3.1.2    Vibration-based condition monitoring**

In general, vibration-based condition monitoring is by far the most prevalent and widely used method, largely because of its ease of instrumentation and its reliable response to damage development (Randall, 2011). First, the majority of all vibration signal processing techniques has some requirement of stationarity. In most cases, stationarity in time is required for harmonic frequencies such that spectrum-based approaches are not invalidated because of frequency smearing. Wind turbines, however,

are far from stationary machines because the wind dictates the rotation speed of the rotor. This speed fluctuation leads to time-varying harmonic frequencies of the vibration sources, such as gears, shafts, or bearing. Knowledge of the speed is therefore crucial for many signal processing methods because this speed fluctuation needs to be compensated or considered. A common, accurate, and reliable way to gain this speed information is through the installation of an angle encoder or tachometer on one of the rotating shafts in the gearbox. An alternative is to estimate the instantaneous angular speed directly from the vibration

signal itself. An overview and comparison of the state of the art in vibration-based rotation speed estimation can be found in Peeters et al. (2019); Leclère et al. (2016).

From a statistical point of view, gear vibration signals are considered deterministic because the gears are locked in place and do not exhibit random slippage like bearings. On the other hand, bearing vibrations are regarded as stochastic in nature because of the random slippage of the roller elements, and they are normally characterized as being second-order cyclosta-

tionary, meaning they have a periodic autocorrelation. This distinction in statistical characteristics provides an opportunity for signal processing methods to separate gear from bearing signals; therefore, a common follow-up step to angular resampling is employing a signal separation technique, such as discrete/random separation (Antoni and Randall, 2004b), self-adaptive noise cancellation (Antoni and Randall, 2004a), linear prediction filtering (Sawalhi and Randall, 2004), the (generalized) time synchronous average (Abboud et al., 2017, 2016), or cepstrum editing (Peeters et al., 2018). Alternative signal separation methods—such as (ensemble) empirical mode decomposition (Huang et al., 1998; Wu and Huang, 2009), principal component analysis, or variational mode decomposition(Dragomiretskiy and Zosso, 2013)—can isolate signal subspaces in the vibration, but these subspaces are typically not guaranteed to have any relevance to mechanical components because they do not employ any prior physical knowledge.

After preprocessing the vibration signal, the last step involves the identification of potential faults. Current practice in condition monitoring systems often revolves around tracking time-domain statistical indicators (de Azevedo et al., 2016; Lu et al., 2009; Tchakoua et al., 2014). Examples of some commonly used time-domain indicators are given in Ali et al. (2018); Zhu et al. (2014); D'Elia et al. (2015); Večeř et al. (2005); Rai and Upadhyay (2016); Sharma and Parey (2016); Decker et al. (1994); Zakrajsek et al. (1993); Bozchalooi and Liang (2007). These time indicators can all be used to characterize trends in measured vibration signals.

The main reason why spectral methods are so popular in condition monitoring is that they allow for not only fault detection but also fault diagnosis. For gears, approaches for fault detection usually focus more on tracking the amplitudes of harmonics and sidebands in the spectrum or the cepstrum, whereas for bearings more cyclostationarity-based methods tend to be employed. Nonetheless, for gear faults it is also recommended to look at the cyclostationary signature of a signal because distributed gear faults can significantly impact the modulation of the deterministic gear signals. A local fault on one of the gear teeth will introduce low-level sidebands in the spectrum, whereas distributed gear damage exhibits higher-level sidebands.

The presence of modulation sidebands is also the main reason why cepstrum-based techniques are popular for gear diagnostics. Because the cepstrum groups together equally spaced harmonics, it provides a very effective means to track the average amplitude of the sidebands. A summary of methods applied for the diagnosis of a faulty gearbox is presented in Sheng (2012).

The most popular approach for the analysis of second-order cyclostationary signals (and, correspondingly, for bearing diagnostics) is envelope analysis. The envelope of a signal is considered to be any function that "encloses" the energy variation in the signal. By taking the modulus of the analytic version of the signal, obtained through the Hilbert transform, the envelope time waveform can be found (given that the envelope frequency of interest respects Bedrosian's theorem (Bedrosian, 1963)). Usually the envelope is squared (effectively done by multiplying the analytic signal with its conjugate) before taking the Fourier transform to analyze its envelope spectrum (Ho and Randall, 2000).

### 3.1.3 Acoustic emissions condition monitoring

To detect strong nonstationary signals such as sudden crack propagation, using acoustic emissions could be a suitable solution. This technology has already been proven for crack detection on pressure tanks, and its applicability for monitoring of rotating components is being researched with promising results. For example, rolling bearings have been exposed to critical operating

conditions, such as high-friction lubrication regimes, overloading, or high angular accelerations, which were successfully detected and differentiated using an acoustic emissions-based detection scheme (Cornel et al., 2018). Further, acoustic emissions have been successfully used for the detection of subsurface cracks in bearings with a response time up to 55% earlier than classic vibration-based detection. Further investigations are being carried out to assess the economic value of this earlier response. Nonetheless, there are still several challenges to be addressed, such as filtering ambient noise or differentiation between individual and overlapping signals of individual components, such as bearings, gears, and couplings.

A further complex but promising challenge of bearing condition monitoring is linking different data acquisition techniques, such as monitoring acoustic emissions and electrical effects or using SCADA data (de Azevedo et al., 2016). To overcome this and the aforementioned challenges, two key aspects need to be addressed in further research:

– Fault diagnosis algorithm: the distinct feature extraction of specific fault mechanisms in real-world applications in non-stationary operating conditions, including the integration between condition monitoring systems; the estimation of the RUL; and new methods of signal analysis, such as machine learning

– Sensor selection and placement: the distinct, transferable description of the influence of fault mechanism specific components inside a complex mechanical system on the system's vibration behavior.

Further, each gearbox has a different behavior; therefore, the first step to reach dynamic system condition monitoring is to obtain reliable characteristic information from the system behavior.

## 3.2   Remaining useful life

The term "consumed life" of a component is a metric that can be calculated based on existing rating methods by using actual loads (whereas design loads are used in the development phase), while "remaining life" is the best estimation of how long a component will survive which requires statistical prediction models. The consumed life calculation of drivetrain mechanical components, such as bearings and gears, requires the analysis of measurements from the field, such as gearbox temperatures, bearing vibration measurements, and wind turbine operational measurements. Such measurements can be continuous online recordings or measurements taken during inspections such as oil quality sampling. Typical indicators for the consumption of the life of the gearbox are:

– The number of particles in the gearbox oil per time duration or the particulates in the grease for greased bearings (Feng et al., 2013)

– The size of the particles in the oil

– The frequency of the oil temperature or bearing temperature excursions above a threshold (Feng et al., 2013)

– Changes in vibration spectrum signatures.

The distribution of particulates in the oil or grease can be analyzed in a laboratory to identify the chemicals present and thereby their source to determine their origin from the bearings or gears.

With advanced multibody simulation tools, it is also possible to model all elements of the drivetrain fully coupled to the aeroelastic interactions of the rotor (Gallego-Calderon and Natarajan, 2015; Gallego-Calderon et al., 2017; Wang et al., 2020) and mounted on a flexible tower. In such a software tool, the drivetrain is subject to continuous wind-driven excitation and grid-driven events. If the response of the drivetrain in terms of the bearing displacements and shaft loads is validated with the physical turbine during different operating conditions, then the software tool can be used to track fatigue damage consumption in the drivetrain by supplying it with measured operating conditions from the physical turbine. This also requires that specific drivetrain component failure modes are tracked in the simulation, such as the occurrences of bearing roller sliding (Dabrowski and Natarajan, 2017) or the vibration excitation of different components. Monitoring fatigue damage growth also provides knowledge of the remaining useful life of the drivetrain if the design life of the components of the drivetrain are known (Doner, 2020). Monitoring systems and modeling tools are also increasingly being integrated with supply chain management systems to reduce downtime and O&M costs (Nordmark and Boyeye, 2021).

Moreover, the inverse methods can also be employed to estimate the loads on drivetrain components from SCADA and vibration measurements and can be used to estimate the component fatigue damage and RUL (Mehlan et al., 2021). Dynamic models of the drivetrain might be needed in some cases if not enough data or measurements are available that can be developed provided that basic information about the drivetrain (e.g., geometry, bearing types, and gear teeth numbers) is known (van Binsbergen et al., 2021).

## 4  Lifetime extension

Once a turbine reaches its 20-year design life, there are, in principle, three options that can be considered by the operator: decommissioning, full repowering, or lifetime extension (Tartt et al., 2021). Full repowering and lifetime extension can accomplish the goal of extending the service life of existing wind turbines. The main benefit is to increase returns on investments and reduce the levelized cost of energy for wind power. Full repowering refers to the dismantling of old wind turbines and replacing them with new ones. Lifetime extension, on the other hand, refers to the assessment of the remaining useful life and possible turbine components upgrades while keeping the turbine hub height, size, or plant layout unchanged. Partial repowering or refurbishment (Topham and McMillan, 2017) is another term used in the industry for replacing and upgrading the components; this can be considered an option in the lifetime extension process.

Whether and how to conduct lifetime extension is a complex decision-making process and depends on many factors, including technical, economical, and legal (Ziegler et al., 2018). The prerequisite is that the turbine's structural integrity—e.g., foundation, tower, nacelle, and hub—has been assessed and can be safely used throughout the expected turbine lifetime extension span. Sometimes these assessments also includes blades, which, if not strong enough, could be upgraded as well. Specific to drivetrain components, typical practices are replacing them with newer products—e.g., gearboxes, main bearings, or generators—which have improved performance and reliability. On the other hand, from the drivetrain research-and-development perspective, some opportunities lie in reliability assessment and driving events identification, which can benefit from fault diagnostics and RUL prediction research conducted to support wind plant O&M. The expected outputs are more

accurate and reliable drivetrain component integrity assessment based on historical data or inspection as well as prediction throughout the planned lifetime extension period. Should the evaluation be positive with an acceptable confidence level, the drivetrain components might not be replaced, as is currently practiced now, and additional costs of the turbine lifetime extension can be reduced.

5    Tartt et al. (2021) investigated the lifetime extension practices in other industries and proposed a methodology for wind turbine drivetrains. The industry's experience with lifetime extension is still limited. Also, different markets—e.g., Europe or the United States—might require different strategies. There are a few uncertainty concerns: i) technically, how trustworthy is the integrity assessment of the structural components; ii) economically, how the assumed future electricity prices might hold true; and iii) legally, how related policies might change.

## 5    Decommissioning and recycling

Wind turbines are typically decommissioned and recycled—at least partially—at the end of their service life because of both their salvage value and local legislative requirements, so decommissioning has increasingly become part of the planning process (Jensen, 2019). Detailed discussions on the decommissioning process for land-based and offshore wind turbines, recycling analysis of different turbine subsystems, basic cost analysis, and the environmental impacts can be found in Jensen (2019). The main challenges surrounding the decommissioning process of wind plants are the regulatory framework, the overall planning of the process, the transportation logistics, and the environmental impacts (Topham et al., 2019a).

Studies show that recycling after decommissioning can pay back some of the decommissioning costs (Topham et al., 2019b). The main parts of the drivetrain from the recycling perspective are the bearings, gears, frames, shafts, couplings, windings, cores (generator stator and rotor), permanent magnets, hydraulic cooling systems, and electronics. Windings are made of copper. Cores are made of electrical steel lamination, which is an iron alloy to which silicon is added. Bearings, gears, frames, shafts, and couplings are made of different types of alloy steel. Permanent magnets used in wind turbine PMSGs are often NdFeB magnets. Hydraulics are considered less problematic because the recycling industry is accustomed to handling the related components. Reuse of lubricants is also being explored (SKF, 2021). Electronics are usually difficult to recycle because of their complex material composition. Nearly all waste electronics equipment is currently shredded and further physical processes, including magnetic and electrostatic techniques, are applied to separate different metal fractions. Two main methods are used to recycle these components: shredding and disassembling. The method used depends on the recycled component, the size, and the material. Recycling can be supported by robotics and using optimization models to optimize the process (Rassõlkin et al., 2018). Separate parts of disassembled components are reused if they do not have any damage. The rest are remelted as the same raw material or to a new alloy. The recycling cost analysis of the drivetrain at the end of the life cycle can be performed by using the flowchart shown in Figure 6. The overall recycling process should be economical under the current conditions and raw material prices.

High-volume drivetrain recycling needs to minimize environmental impact (no harmful chemicals/emissions or energy-intensive processes) and have high recovery rates of rare and precious materials while still being economically viable. Recycling

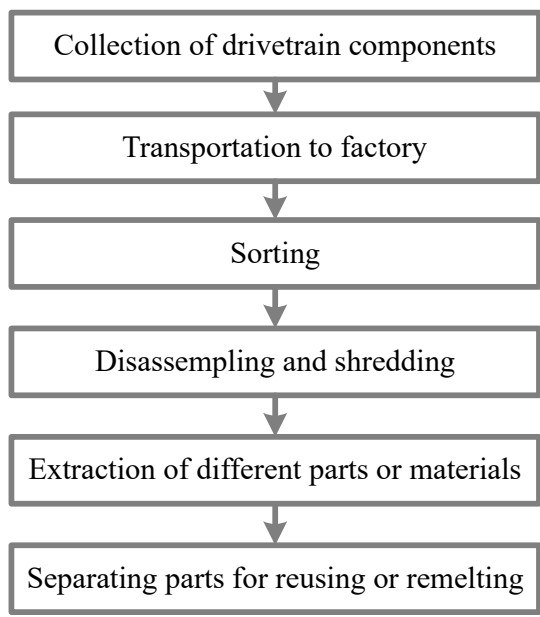

**Figure 6.** Recycling cost analysis of the drivetrain at end of life.

can also be in the design phase. For example, surface-mounted PMSGs compared to interior rotor magnets, have different disassembling procedures. Modular design can also offer a clear advantage in recycling because healthy parts can be separately reused in other components of that type.

Further, the use of recycled material can lead to cheaper new products (Gabhane and Kaddoura, 2017), which, apart from being more environmentally friendly, can be a good motivation for recycling. Recycling and reusing can also contribute to reduce the potential supply chain risk, especially for rare-earth materials (Habib and Wenzel, 2014).

## 6   Emerging areas

### 6.1   Drivetrain in floating turbines

Drivetrains in floating wind turbines are exposed to different dynamic loads than those on bottom-fixed or land-based ones. Apart from the wind loads, the wave-induced motions can affect the drivetrain load responses. The wave-induced motions can have a negative impact on the main bearing fatigue life—particularly on the one carrying axial loads—as highlighted by Nejad et al. (2015). Sethuraman et al. (2014) investigated the effects of the floating wind turbine motion on direct-drive generator air gap integrity and showed that the air gap stability of the generator is more sensitive to magnetic forces if the supporting frame is relatively rigid. They also highlighted the need for air gap management for direct-drive generators on floating platforms.

A recent full-scale experimental field study of a 6-MW drivetrain on a spar floating substructure (Torsvik et al., 2021) indicates that the effect of wave-induced motions might not be as significant as the wind loading on the drivetrain responses,

particularly in larger turbines. A set of strain measurements on the main bearing shows that the effect of wave motions is negligible compared with the tower shadow excitation (Torsvik et al., 2021).

The study by Nejad and Torsvik (2021) investigated the lessons learned in the last 10 years with regard to drivetrains on floating wind turbines. Among others, the study highlighted that the maximum tower top axial acceleration might not be a reasonable limiting factor for the fatigue life of main bearings in floating wind turbines, at least for a spar type floating turbine (Nejad et al., 2019). It also emphasized that a flexible bedplate influences the main bearing and components inside the gearbox, and therefore the coupling effects between the structure and the drivetrain need to be considered in floating wind turbines, particularly in compact design concepts. Given the limited experience with floating wind turbines, however, more research is needed.

## 6.2 Drivetrain and plant consideration

Wakes induced by other turbines in the plant cause increased turbulence and thus can affect the loading on the drivetrain (Roscher et al., 2017). How the wakes are controlled at the plant level will therefore influence the drivetrain design life (van Binsbergen et al., 2020). Several approaches which have recently been proposed to reduce overall plant losses by reducing wake effects employ either some form of static or dynamic induction control or wake steering by yaw control (Andersson et al., 2021). Static induction control aims to reduce the strength of the wake of an upstream machine by changing the pitch of the blades or the rotational speed of the rotor in such a way as to reduce the thrust at the expense of some efficiency but to allow downstream machines to see an increased wind speed so that the combined output of the turbines is increased. However, in practice, little gain is seen using this approach (Bartl and Sætran, 2016). Dynamic induction control seeks to increase wake mixing and thus reduce plant losses by periodically pitching the blades during each rotor rotation (Alexis Frederik et al., 2020). Wake steering takes a different approach whereby an upstream turbine is deliberately yawed out of the prevailing wind direction (again slightly reducing turbine efficiency) with the aim of ensuring downstream machines are out of its wake, thus increasing their output and increasing overall plant efficiency (Fleming et al., 2017). Wake steering typically does not have large effects on the drivetrain of the upstream turbine, whereas an increase in the yaw angle would lead to an increase in the drivetrain load variation of the downstream turbine (van Binsbergen et al., 2020). In contrast, increases in blade pitch angle associated with static induction control would reduce the standard deviations of the drivetrain dynamic response of the upstream wind turbine, whereas it would not significantly affect the standard deviations in the downstream turbine (van Binsbergen et al., 2020).

Based on the study performed by van Binsbergen et al. (2020), the wake impact on the downstream turbine can reduce the lifetime of the first main bearing of the drivetrain by 17%, and it can reduce the turbine power intake by 30%. The mitigation of loads on the drivetrain of the wind turbine and an increase of power capture at the turbine level is addressed in the literature on turbine control by optimizing the generator torque, blade pitch, and yaw steering controls (as shown in, *e.g.*, van Binsbergen et al. (2020) and Fleming et al. (2013)). Optimized wind power plant management by considering the influence of wakes was recently studied by Andersson and Imsland (2020).

The optimized plant control design to maximize the wind plant power intake and simultaneously minimize the degradation of the drivetrain components influenced by wake loads is still an open research problem. The latter calls for high-fidelity models

of wake flow, wake loads on the drivetrain of downstream turbines and the drivetrain system to study the drivetrain load effects and responses, and, finally, the derivation of sufficient drivetrain lifetime-related constraints that can be integrated into the plant stochastic model predictive control design framework. From this perspective, the additional role of wind plant control is to distribute accumulated fatigue evenly over the drivetrains of different turbines to improve the reliability of the plant.

## 6.3 Drivetrain and digitalization

The use of digital technologies and digitized data can support wind turbine drivetrain analyses at both the system and component levels during the life cycle—including design, installation, and O&M—and lifetime extension from various aspects, namely, by receiving and transmitting real-time data through the data acquisition and transmission layers; by storing and processing data through the platform layer, and, finally, by decision support through the application layer (An et al., 2021). To this purpose, digitalization targets the sensors and actuators installed on the drivetrain and the other turbine systems—such as the site network, servers, and even smart phones (André et al., 2021)—that are connected to the turbine and plant's control and monitoring systems to improve reliability, availability, quality of service, and user experiences.

Digitalization enables digital twin models that can support the drivetrain's design and operation (Moghadam and Nejad, 2022). Digital twin in this context includes sensors (data collection), models (dynamic and degradation models), and decision support platform (e.g., estimation of remaining useful life) (Johansen and Nejad, 2019). The application of computationally inexpensive digital twin models for the predictive maintenance of drivetrain system components by monitoring the remaining useful lifetime of the critical components (*e.g.*, the gears of the gearbox) in real time has been proposed in recent literature (Moghadam et al., 2021). Using the existing physical models, integrating them in a unified framework, applying signal processing techniques to estimate these models and model inputs in real time from available measurements, optimizing the data streaming between models, using continuous processing architectures, and using statistical approaches and stochastic modeling techniques to model and mitigate the impact of uncertainties are the tasks under the umbrella of a digital twin. The risk associated with digital twin implementation, specially in terms of wrong decision support, should also be considered (Ibrion et al., 2019). In general, challenges of digital twin models mainly arise from (Moghadam et al., 2021):

– Finding the minimum or appropriate model fidelity required to capture the dynamics of the component for different operation and failure modes for a wide range of drivetrain components

– Optimizing data streaming between models, data processing algorithms, and continuously processing architectures to deal with the real-time aspects of digital twin models

– Developing a preprocessing stage to model and mitigate the different sources of uncertainty and to verify and ensure time synchronicity when the data come from different sources and when dealing with the different sample rates.

To overcome these challenges, merging edge computing with the Internet of things (IoT), as shown in Figure 7, can play a significant role. Using distributed computing algorithms supported by edge computing data handling architectures (*e.g.*, fog computing) has been recently adapted from computer science for wind plant control and monitoring systems to significantly

reduce the computational burden of implementing digital twin models' complex algorithms to monitor drivetrain components in future wind turbine O&M analyses (Verstraeten et al., 2019). By using fog computing, it is possible to break the digital twin into simple subproblems with less computational complexity, and each one can be executed on a fog. IoT can provide real-time access to data in each network node with the possibility of using the processing and storage capacities of the nodes for control and monitoring purposes, and not all data in fogs need to be shared with other fogs or even the cloud. Cybersecurity frameworks should be deployed at the communication network and the computing modules to guarantee the integrity, authenticity, confidentiality, and availability of the data while they are being transmitted over the network nodes. Because future wind plants' control and monitoring systems will need to handle more voluminous, heterogeneous data and distributed features, but storage and processing capacities are limited, the IoT can significantly improve turbine control and monitoring.

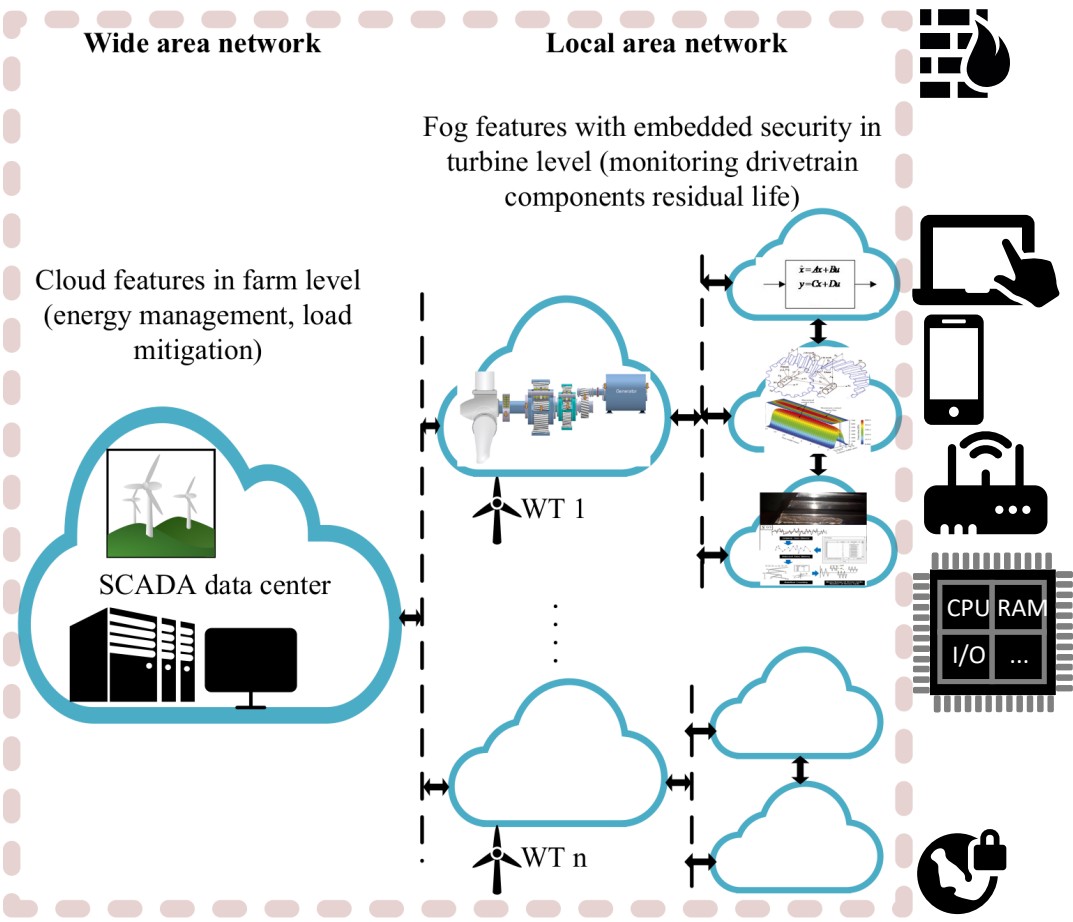

**Figure 7.** Drivetrain system monitoring by means of digital twin models and distributed data management architectures and algorithms.

## 7   Summary and concluding remarks

This paper presented the state of the art and future development trends for wind turbine drivetrain technologies from the life-cycle perspective. Lighter and more compact design concepts are the most cost-effective option, particularly for large offshore turbines. As a result, the integration of electromechanical systems together with the main bearing has been a recent trend. For land-based applications, geared drivetrains seem to be the dominant technology; however, the race between geared or not is still ongoing for offshore, with medium-speed (or hybrid concept) or even high-speed geared drivetrains under development for large offshore wind turbines. The extensive research and development of gearbox and bearing design has mainly been on improving reliability by better understanding the failure modes in operation, on one hand and, on the other hand, on the development of new technologies, e.g., plain bearings. The medium-speed concepts have shown to be a promising compromise between high-speed and direct-drive designs, whereas high-speed, multistage gearboxes have also been a research focus in recent years. For offshore turbine generators, there has been a recent increasing trend toward direct-drive systems, PMSGs with full-power converter systems rather than DFIGs with partial-power converter systems. Superconducting generators are also seen as an attractive alternative to PMSGs because of the large amount of rare-earth materials needed for PMSGs.

Given the criticality of drivetrain components in wind turbines, condition-based maintenance is seen as an essential element, at least for offshore and large turbines. In newer and larger turbines, greater than 2.5 MW or 3 MW, most if not all have dedicated condition monitoring systems, which typically monitor the gearbox, main bearings, and generator. Their outputs can be used to support anomaly detection, fault diagnostics, and prognostics models. These models can be data-driven, physics-based, or hybrid, integrating both data and physics-domain models. In the data domain, particularly for SCADA-based condition monitoring, machine learning and artificial intelligence technologies are being actively investigated for wind turbine applications.

Condition monitoring technologies deployed for wind turbine drivetrains are generally good at fault diagnostics, especially for high-speed components. The performance of various solutions in terms of prognostics still needs to improve, which presents an opportunity to bridge progress made by the research community. On the other hand, the industry has long been eager to obtain accurate predictions of component remaining useful life, which is one objective of typical fault prognostics. Among the various drivetrain subcomponents, bearing faults have shown to be prevalent and have been actively investigated by both industry and researchers.

As the industry moves farther offshore and into deeper water, increasing numbers of floating wind turbines are expected to be used. The early lessons highlight differences in terms of dynamic behavior and life of the drivetrain in floating wind turbines compared with fixed ones, especially for the main bearings. As offshore turbine sizes are increasing, the component flexibility and potential dynamic coupling effects should not be overlooked during design modeling and analysis.

Another emerging area of research for drivetrains is digitalization. Apart from data processing and digital models, data handling, ownership, security, communication, and transfer at the scale of large wind plants are also interesting challenges. The use of digital twins for condition monitoring is also a recent research direction.

This article also illustrated that the drivetrains in wind turbines are very multidisciplinary objects in all stages of their life cycles—from design to operation, to lifetime extension, to end of service and recycling. This calls for more interdisciplinary research and collaborations to improve wind turbine drivetrain reliability and availability with the main aim to reduce the cost of energy over time.

5 *Competing interests.* No competing interests are present

*Acknowledgements.* This paper has been prepared by the Drivetrain Technical Committee (DTC) at the European Academy of Wind Energy (EAWE) over period of 2020-2021. The authors appreciate fruitful discussions in DTC and acknowledge EAWE for facilitating this forum for the wind research community.

This work was also authored in part by the National Renewable Energy Laboratory, operated by Alliance for Sustainable Energy, LLC, for 10 the U.S. Department of Energy (DOE) under Contract No. DE-AC36-08GO28308. Funding provided by U.S. Department of Energy Office of Energy Efficiency and Wind Energy Technologies Office. The views expressed herein do not necessarily represent the views of the DOE or the U.S. Government. The U.S. Government retains and the publisher, by accepting the article for publication, acknowledges that the U.S. Government retains a nonexclusive, paid-up, irrevocable, worldwide license to publish or reproduce the published form of this work, or allow others to do so, for U.S. Government purposes.

15 E. Hart is funded by a Brunel Fellowship from the Royal Commission for the Exhibition of 1851.

Pieter-Jan Daems, Timothy Verstraeten, Cédric Peeters, and Jan Helsen received funding from the Flemish Government (AI Research Program). They would like to acknowledge FWO (Fonds Wetenschappelijk Onderzoek) for their support through the SB grant of Timothy Verstraeten (#1S47617N) and post-doctoral grant of Cédric Peeters (#1282221N). They would also like to acknowledge VLAIO for the support through the SIM MaDurOS program project SBO MaSiWEC (H.B.C.2017.0606) and Blauwe Cluster ICON project Supersized 4.0.

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
