# Peer review of "Wind turbine drivetrains: state-of-the-art technologies and future development trends"

_Wind Energy Science, 2021_

## Author Response (AR1)

Authors would like to thank reviewers for their constructive comments. All comments have been addressed and the paper is revised accordingly.

**Reviewer 1:**

**Comment 1:** This is a useful review paper covering the different elements of the wind turbine drive train, including condition monitoring.  It is reasonably structured, although the modelling and analysis section (2.5) really only applies to the gearbox.  As such it could be moved into section 2.2.
**Response 1:** Section 2.2 covers the design trends, not limited to modelling, while section 2.5 is dedicated to modelling only. We agree that 2.5 in its current form applies to the gearbox. We have updated and included modelling of generator in the revised version.

**Comment 2:** Some text corrections/clarifications are required: p5 line 9 - should "epicyclic systems" be stages ?
**Response 2:** this sentence has been revised as: With multistage gearboxes using four or more planetary gear systems…

**Comment 3:** 2.2 This section is missing important discussion of gearbox casing distortion under load, and the influence on gear and bearing loading
**Response 3:** This is now included in revised version.

**Comment 4:** p6 line 7 "It aims to achieve a trade-off between generator size and maintenance effort." - unclear
**Response 4:** This sentence has been updated.

**Comment 5:** p6 line 27  should this be simply generators ? not specifically PM machines
**Response 5:** Agreed and corrected.

**Comment 6:** p6 line 32  compensating gearbox reliability needs to be mentioned to provide the correct context
**Response 6:** Agreed and updated.

**Comment 7:** p7 line 11 speed should be shaft speed
**Response 7:** This has been changed.

**Comment 8:** p8 line 3 power should be converter with power
**Response 8:** This has been changed.

**Comment 9:** p8 line 16  the nature of the energy storage should be mentioned
**Response 9:** The energy storage is lithium-ion battery. It is now mentioned in the paper.

**Comment 10:** p8 line 20 the term "rotating" is unclear
**Response 10:** It is now corrected to 'the rotating speed control'

**Comment 11:** p9 line 2 the term "grid-supporting mode control" needs to be more fully described and converter topology referenced
**Response 11:** It is now corrected to 'grid-following mode control'

**Comment 12:** p9 line 7 an outline description referencing converter topology and control is required for grid-forming mode control
**Response 12:** More elaboration is now added

**Comment 13:** p9 line 14 "converter choice of cooling system" needs to be rephrased
**Response 13:** It is now changed to 'The type of cooling system chosen for the converter'

**Comment 14:** p12 lines 21 to 25 seems to be written assuming offshore wind - onshore and offshore O&M costs should be distinguished
**Response 14:** This has been clarified and additional references added.

**Comment 15:** p13 line 8 'sampled' is misleading - the sampling rate in generally much higher with averaged results saved every 10 minutes
**Response 15:** This has been rewritten to make it clearer.

**Comment 16:** Section 3.1.1 changes to power curves is a powerful method for identifying insipient faults - Gaussian Process models have recently been found to very effective at this. This emerging approach should be discussed and referenced
**Response 16:** A discussion on the use of GP models has been added under 3.1.1 (iii) NBM.

**Comment 17:** p20 line 15 spar support structures are reasonably rigid, this conclusion will not apply too all floating wind - this should be made clear
**Response 17:** we have modified the sentence to make it clear that the case study turbine was on a spar support structure. Regarding the "spar support structures are reasonably rigid" we would like to highlight that the wave induced motion of the case study spar was the highest among the TLP, and two other semi-submersible turbines - please see: https://doi.org/10.1016/j.marstruc.2015.03.006

**Comment 18:** p20 line 25 "plant" should be wind farm
**Response 18:** Our language editor insisting on the word "plant". They view the proper term as wind plant. A "farm" they view as colloquial term.

**Comment 19:** p20 line 27 the term wake steering needs to be explained and its pros and cons discussed
**Response 19:** This paragraph has been expanded to explain wake steering, wake mixing and induction control and to more clearly relate these to impact on the drivetrain.

%%%%%%%%%%%%%%%%%%%%%%%%%%%%%%%%%%%%%%%%%%%%%%%%%%%%%%%%

**Reviewer 2:**

Authors would like to thank reviewers for their constructive comments. All comments have been addressed and the paper is revised accordingly.

**Comment 1:** The paper gives an interesting and comprehensive overview of technologies and research related to modern wind power drivetrains. It could elaborate a bit more on drivers for technical developments in the introduction, e.g. how are different regions demanding different solutions (US, EU, China, India, offshore, ...)? - how is logistics impacting future designs? - how will increased need for availability (higher capacity factor) impact future designs?
**Response 1:** Following discussion is now added to introduction: "It is interesting to highlight that the technological drivers for drivetrain are not necessarily the same as other elements of wind turbine. For instance, for towers there are site specific solutions, depending on specific wind conditions or specific site characteristic, which effect the cost considerably. However for drivetrain, the number or size of components are cost drivers, not the site or region specific characteristics.
It is possible to design the drivetrain with respect to the logistic costs meaning modular design which can be handled by crane, reducing transportation or marine operation cost in case of offshore wind turbines, although such technology is yet to be proven. In terms of drivetrain availability for future design, the development is not only on the quality and manufacturing, but also on the service and operational monitoring".

Hereby specific suggestions for the text:

**Comment 2:** fig 1 => I would recommend to have in the 1st picture of a paper with a state-of-the-art overview a more typical setup of the drive train => for the picture of a wind turbine with a gearbox, I recommend to use a classical 3- or 4-point suspension setup, not an axle-pin design since this is less common; same for the gearbox picture => I would recommend to use a picture from a typical 3-stage high speed gearbox (from a 3- or 4-point suspension)
**Response 2:** Figure 1 is modified.

**Comment 3:** 2.2 - 25 - it might be good to reference VDMA 23904 as a method to calculate reliability of wind turbine gearboxes
**Response 3:** The existing sentence in 2.2 has been modified "…the reliability of the gearbox is the product of the reliability of all the failure modes for which there exists a reliability calculation as described in Verband Deutscher Maschinen- und Anlagenbau 23904 and IEC Technical Specification 61400-4-1."

**Comment 4:** 2.2 - 34 - "life-limited only by wear" - it is good to add "which means not driven by load dependent fatigue"
**Response 4:** The existing sentence in 2.2 has been modified "…are only life-limited by wear rather than determined by load-dependent rolling contact fatigue, although…."

**Comment 5:** 2.5 - quite a lot of details are described on gear modelling - it would be good to indicate that a lot of gear simulation methods are to predict vibration and consequent noise behaviour,

whereas the focus in this overview is mainly on calculating stiffness and stresses during wind turbine load conditions (to assess durability of designs)

**Response 5:** We agree that 2.5 in its current form applies to the gearbox dynamic modelling only. We have expanded this section and included modelling of other components in the revised version.

**Comment 6:** p 12 - line 27 - it would be good to explain "digital twin" (or put a reference) since this is a quite broad term

**Response 6:** A reference to section 6.3, where digital twin is defined, is added.

**Comment 7:** p 13 - line 5 - please explain SCADA (or put a reference)

**Response 7:** The acronym SCADA is explained at the beginning of Section 3 and what the system entails is explained in Section 3.1.1. The first sentence in 3.1.1 has been extended slightly to make it clear that the SCADA data come from sensors on the wind turbine.

**Comment 8:** Table 1 - it would be good to add a reference to the relevant IEC61400 chapter

**Response 8:** Reference to IEC 61400-25-2:2015 has been added.

**Comment 9:** p 17 - 3.3 - it would be good to dinstinguish between "consumed lifetime" and "remaining lifetime" since they are both used. I would recommend to use "consumed lifetime" as a metric that can be calculated based on existing rating methods, by using actual loads (whereas design loads are used in the development phase); "remaining lifetime" should be the best estimation of how long a component will still survive => here it is senseful to include statistical methods in combination with conditional parameters (temperature, vibrations, ...)

**Response 9:** Agreed and included.

**Comment 10:** p17 - it would be good to refer to industrial use cases (e.g. Boeye-ZF, Nordmark-SKF presented at CWD conference 2021)

**Response 10:** Agreed and included.

---

## Editor Decision (ED1)

WES-2021-63

Overall
- The paper is well-written and thorough. It is still missing a bit of substance that really motivates the importance of the paper. Can you further elevate and pull out what are the big challenges / exciting developments and emphasize them further in the abstract, intro and conclusions?
- There are a few typos that a final edit will catch
- The structure of sections 2 through 5 might be revisited. Section 2.5 could be its own section. Sections 4 and 5 could be integrated into section 3 – they dont stand well on their own

Abstract
- The abstract has a bit of redundancy and is a bit lacking in terms of a punchline – consider pulling out a few key highlghts from the paper and emphasizing here. What are the biggest challenges? What are the most significant trends? Innovation game changers?

Introduction
- The added paragraph in the introduction seems to be tangential – it seems like it would fit better in section 2 as it speaks to design drivers
- The intro is still missing a bit of the why. The first paragraph is good – gigantic machinery in new environments (i.e. floating) are challenging the status quo – maybe provide an example or two of what specifically for drivetrains is so challenging? Have there been some instances of serial failures or is there an expectation that moving to 20 MW size turbines poses new challenges? Have digital twins shown potential to unlock potential for O&M cost reductions / significant increases to availability?

Design trends and developments
- The shift towards SGs for offshore is largely driven (to my understand) by the desire to move to direct-drive systems and eliminate gearboxes offshore – is that not the case?
- For figure 1 – it might be better to split into two figures – one showing a geared and on eshowing a direct-drive configuration since the righthand now is mixing elements of each. The middle figures could be expanded with labels pointing to the major components of each
- I agree with the reviewer that section 2.5 still seems out of place. Maybe it should be its own section?

Emerging areas
- Consider adding a short introductory paragraph before section 6.1 explaining why these 3 topics are so important to consider
- In terms of loading from control strategies, there is a good bit of work on the topic from NREL and SNL that would be worth pulling out. There was a lot of work

partricularly looking at main bearing loads… also yaw bearing though the yaw system is not considered in this work
- I would have expected to see a bit more on digitalization and digital twins – perhaps the literature on the topic is not that extensive yet?